# Experimental search for high-temperature ferroelectric perovskites guided by two-step machine learning

Prasanna V. Balachandran [1,3], Benjamin Kowalski[2], Alp Sehirlioglu[2] & Turab Lookman[1]

Experimental search for high-temperature ferroelectric perovskites is a challenging task due to the vast chemical space and lack of predictive guidelines. Here, we demonstrate a two-step machine learning approach to guide experiments in search of $x\mathrm{Bi}[\mathrm{Me}'_y\mathrm{Me}''_{(1-y)}]\mathrm{O}_3$–$(1-x)$ $\mathrm{PbTiO}_3$-based perovskites with high ferroelectric Curie temperature. These involve classification learning to screen for compositions in the perovskite structures, and regression coupled to active learning to identify promising perovskites for synthesis and feedback. The problem is challenging because the search space is vast, spanning ~61,500 compositions and only 167 are experimentally studied. Furthermore, not every composition can be synthesized in the perovskite phase. In this work, we predict $x$, $y$, Me′, and Me″ such that the resulting compositions have both high Curie temperature and form in the perovskite structure. Outcomes from both successful and failed experiments then iteratively refine the machine learning models via an active learning loop. Our approach finds six perovskites out of ten compositions synthesized, including three previously unexplored {Me′Me″} pairs, with $0.2\mathrm{Bi}$ $(\mathrm{Fe}_{0.12}\mathrm{Co}_{0.88})\mathrm{O}_3$–$0.8\mathrm{PbTiO}_3$ showing the highest measured Curie temperature of 898 K among them.

[1] Theoretical Division, Los Alamos National Laboratory, Los Alamos, NM 87545, USA. [2] Department of Materials Science and Engineering, Case Western Reserve University, Cleveland, OH 44106, USA. [3] Present address: Department of Materials Science and Engineering, Department of Mechanical and Aerospace Engineering, University of Virginia, Charlottesville, VA 22904, USA. Correspondence and requests for materials should be addressed to P.V.B. (email: pvb5e@virginia.edu) or to A.S. (email: axs461@case.edu) or to T.L. (email: txl@lanl.gov)

One of the important challenges in the computer-guided accelerated search for new materials is the simultaneous optimization of stability and one or more properties of interest[1,2]. Although the rationale for property optimization can be readily justified (e.g. need for room temperature superconductors, fast ion-conductors, sub-nm size skyrmions etc.), predicting stability is critical because it can inform whether a candidate material can be potentially synthesized in a laboratory setting or not. Furthermore, a quantitative understanding of the stability can put a constraint on the feasible solution space and our ability to predict them can significantly impact the accelerated search for new materials. Traditionally, density functional theory calculations are utilized, where the formation enthalpy and distance from the convex hull are used to determine the stability or metastability of a compound or an alloy[3–8]. While powerful, this approach has limitations in the accelerated search, especially if complex solid-solutions with fractional site occupancies, large supercells, and transition metal oxides with strong electron correlation are involved. The state-of-the-art computational tools[9–11] are most suited to handle stoichiometric compounds.

In contrast, machine learning (ML) approaches that use experimental data are also increasingly utilized for accelerating the search for new materials[12–16]. In these studies, data from both successful and failed experiments are used to train ML models, which in turn can be used to predict whether a new data sample (not present in the training set) can be potentially experimentally synthesized or not. Thus, these methods do not use or strictly require thermodynamic stability data but take advantage of the past experiments to make inference about future experiments. One of the emerging areas in the nascent field of materials informatics is the active learning or adaptive design approach, where the ML models are combined with algorithms that recommend informative experiments (from a vast pool of possible experiments) such that the new data are expected to improve the performance of the ML models in the next iteration[17–19]. Recent demonstrations of these methods to experimentally discover complex organic–inorganic molecules, alloys, and functional oxides are worth mentioning[12–16].

In this paper, we focus on the complex $x\text{Bi}[\text{Me}'_y\text{Me}''_{(1-y)}]\text{O}_3$–$(1-x)\text{PbTiO}_3$ (PT) perovskite solid solutions, where Me′ and Me″ are di-, tri-, tetra- or pentavalent cations that occupy the octahedral site of the perovskite lattice (shown in Fig. 1) and the resulting solid solution is charge neutral. These are candidate materials for high-temperature ferroelectric applications because of their large piezoelectric and electromechanical coupling coefficients[20]. Our survey of the periodic table identified a total of 23 cations that can occupy the Me′ and Me″ octahedral sites in the perovskite lattice. Further, these 23 cations can be distinctly combined to yield a total of 75 {Me′Me″} possible cation pairs and only 13 such pairs are experimentally explored. We also allow $x$ to vary from 0.05 to 0.85 in steps of 0.05 and $y$ to vary from 0.1 to 0.9 in steps of 0.01 for constraining our composition space. Overall, we identify 61,506 unique chemical compositions out of which only 167 are experimentally studied, representing only 0.28% of the search space. In the remaining unexplored chemical and composition space, it is unclear how many candidate high-temperature ferroelectric perovskites can and do exist. This is a challenging question because rules governing formation of high-temperature ferroelectric perovskite phases in a complex multinary phase space are not known a priori. Traditionally, time-consuming and expensive trial-and-error or intuition-driven experimental approaches are used to uncover the composition–structure–property relationships, which is a non-trivial task. The overarching research problem is schematically shown in Fig. 1.

Here, we demonstrate a materials design approach driven by ML and active learning methods to simultaneously predict $x$, $y$, Me′, and Me″ such that the new candidate solid solutions are expected to (i) form in a perovskite structure (with at least 95% phase purity) and (ii) also have high ferroelectric Curie temperature ($T_C$). The novelty of our ML approach lies in the integration of classification learning with regression methods to constrain the search space of possible perovskites so that only promising compositions are recommended for experimental synthesis, characterization, and feedback. As a result, we build two independent ML models, one for classification learning and the other for regression. The data for training the ML models are taken from the published experimental literature[21–45]. While the classification learning models allow us to screen for candidate chemical compositions that can have perovskite structure, the

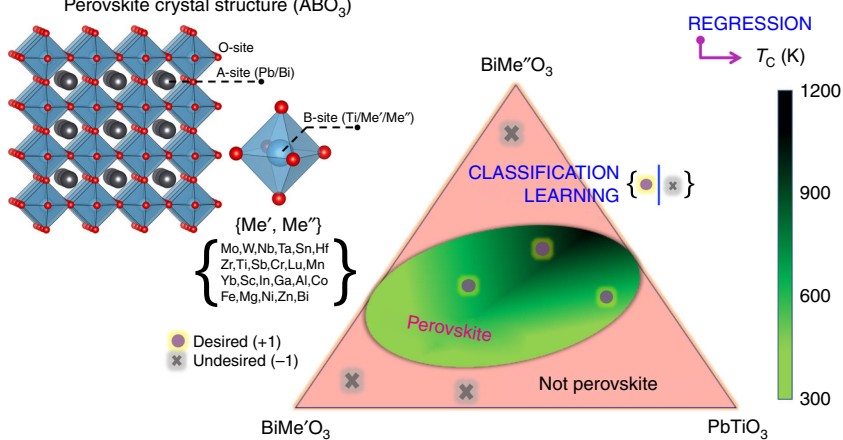

**Fig. 1** Materials design challenges in the search for high-temperature ferroelectric perovskites. The phase space of $x\text{Bi}[\text{Me}'_y\text{Me}''_{(1-y)}]\text{O}_3$–$(1-x)$PT solid solutions contains a total of 61,500 compositions and only about 0.28% of them are experimentally investigated. A priori, the stability field of the desired high-temperature ferroelectric perovskite phase is not known. Traditionally, trial-and-error or intuition-driven experimental approaches are used to discover new high-$T_C$ compositions, which is non-trivial and can be time-consuming and costly. We demonstrate a two-step ML approach that can simultaneously optimize $x$, $y$, Me′, and Me″ such that the resulting solid solution will not only have a high-$T_C$, but form in the perovskite crystal structure. Classification learning methods identify and isolate desired regions in the phase diagram, where the candidate perovskite phases are expected to exist. Regression methods, on the other hand, predict the ferroelectric $T_C$ for the candidate perovskites. Active learning methods recommend promising high-$T_C$ perovskite compositions for synthesis. Experiments validate the ML predictions and provide feedback for further model improvement

regression models then predict the $T_C$ for the candidate perovskite solid solutions. We also use our recently developed active learning (or adaptive design) approach based on efficient global optimization (EGO)[13,17] to recommend promising compositions for experimental synthesis and characterization (details of EGO algorithm are discussed in the Regression and Active Learning section). We employ the conventional powder processing technique for synthesis, which allows us to validate one of the important questions of perovskite stability under the driving force of temperature. One can also, in principle, experimentally stabilize the perovskite phase by means of pressure, but we focus only on the temperature. We run our iterative loop for a total of five times, which resulted in investigating ten new compositions and we discovered six new perovskites, with $0.2Bi(Fe_{0.12}Co_{0.88})O_3–0.8PT$ having the highest measured $T_C$ of 898 K among them. We also identify three novel {Me′Me″} pairs, namely {FeCo}, {CoAl}, and {NiSn}, that are not explored in the literature. Since the ceramic processing route that we have employed is akin to what industry uses in scaled-up production, the novel composition spaces identified in this work can potentially impact the development of functional materials for high-temperature applications such as piezoelectric actuators[46,47]. The current work is also a departure from those reported in the literature[12,13,15,44] in the following ways. First, in terms of the ML approach, we setup our materials design problem as two-step learning to sequentially guide experiments, where a classification learning model downselects promising candidates in the desired crystal structure and a regression method coupled to active learning recommends promising compositions for experimental synthesis. The new predictions are expected to simultaneously satisfy two criteria that are crucial for practical applications: first, they are expected to form in the desired perovskite crystal structure and second, they must also have a promising property, i.e., high ferroelectric Curie temperature ($T_C$). The potential of ML to constrain the search space has major implications in rationally guiding experiments towards promising materials with targeted properties. Previous studies considered only either the classification learning or regression methods for design[12,13,15]. Second, in terms of the materials class, the focus is on complex functional oxides that are synthesized via conventional powder processing routes. However, previous demonstrations of classification learning with feedback from experiments have been on complex organic−inorganic molecules[12,15]. Third, we use our strategy to sequentially guide 1−3 experiments at each iteration step, in contrast to the batch experiments of Duros et al.[15], who performed ten new experiments at each iteration step for validation and feedback.

## Results

The need for a two-step learning strategy, especially the classification learning for constraining the perovskite composition space, was motivated by our first prediction on the $0.5Bi(In_{0.36}Sc_{0.64})O_3$-$0.5PT$ composition. We predicted this solely on the basis of regression and active learning under the assumption that all compositions in the constrained $xBi[Me′_yMe″_{(1−y)}]O_3−(1−x)$ PT search space can be potentially synthesized in the perovskite phase. We did not use classification learning to evaluate whether this composition will form in the perovskite structure or not. This naive strategy proved to be insufficient, because the X-ray diffraction (XRD) measurements revealed the presence of secondary phases. This indicated that the composition did not form in a single-phase perovskite structure and furthermore, these secondary phases are deleterious for high-temperature applications. This failed attempt led us to reformulate our approach and we constructed a new dataset for classification learning, whose primary objective is to identify promising regions in the $xBi[Me′_yMe″_{(1−y)}]O_3−(1−x)PT$ search space that are expected to form in the pure perovskite phase. The reformulation resulted in the development of a two-step ML strategy, which we show in Fig. 2. The synergistic effect of employing classification learning, regression, active learning, experimental validation, and feedback in the discovery of high-$T_C$ ferroelectric perovskites are discussed below.

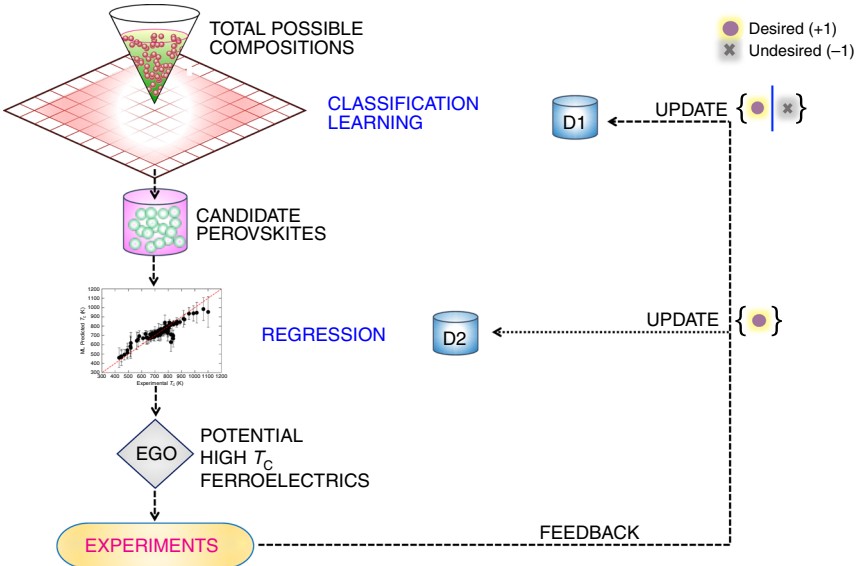

**Fig. 2** Two-step machine learning strategy for sequentially guiding experiments. We construct two independent datasets (D1 and D2) from surveying the experimental data in the literature, one for building classification learning and the other for regression models. The role of classification learning is to screen for compositions that can be synthesized in the perovskite phase. Compositions that pass the classification learning screen are referred to as the "candidate perovskites". The regression models then predict the $T_C$ with associated uncertainties ($\sigma$) for the candidate perovskites. We then use efficient global optimization (EGO)[61] to identify promising high-$T_C$ candidates for experiments. Outcome from both successful and failed experiments provides feedback for classification learning. On the other hand, we only use the outcome from successful experiment to update the regression ML models. We iterate our design loop for a total of five times

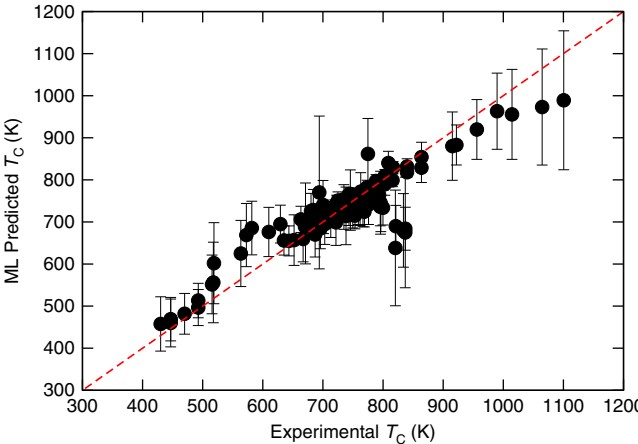

**Fig. 3** Performance of SVR$_{rbf}$ on the training dataset. Experimental ($x$-axis) vs. ML predicted ($y$-axis) $T_C$ for the initial dataset of 117 compounds. The error bars are the standard deviation from the predictions of 100 SVR$_{rbf}$ ML models on the dataset. The dashed red line is the $x = y$ line, where the predictions from SVR$_{rbf}$ models and experimental $T_C$ are equal

**Classification learning**. We constructed an initial dataset of 167 polycrystalline ceramic samples taken from the published literature. The compositions in the dataset were synthesized using either the conventional powder processing, solid state reaction methods, or mixed oxide processing routes, with some differences in the calcination and sintering protocols. Thus, we have a heterogeneous dataset as it is constructed from diverse sources. The phase purity of the polycrystalline ceramics in our training set was determined using the XRD measurements. Compositions that were revealed to be at least 95% phase pure in the perovskite structure were labeled as "+1" (or desired) in our dataset and we do not explicitly distinguish between tetragonal, rhombohedral, or monoclinic perovskite crystal structures. Similarly, those that had secondary phases were labeled as "−1" (or undesired). Our training set contained 107 and 60 compositions with +1 and −1 labels, respectively.

We represent each composition in our dataset using five features, namely tolerance factor, valence electron number, Martynov−Batsanov electronegativity, ideal bond lengths, and Mendeleev number[48–51]. Recently, Pilania et al.[52] showed the relative importance of these features in classifying the formability of ABO$_3$ compounds in perovskite crystal structure, and therefore, we use them in this work. Since every composition is a solid solution, we used the weighted-fractions of the relative proportions of Bi, Pb, Me′, Me″, and Ti in the solid solution to calculate the feature values. For instance, the tolerance factor ($t_f$) for (say) a $x$Bi(Me′Me″)O$_3$ − $(1 − x)$PT solid solution is calculated using the formula,

$$t_{f,\text{solid solution}} = \{x \times t_{f,\text{Bi(Me′Me″)O}_3}\} + \{(1 − x) \times t_{f,\text{PT}}\}, \quad (1)$$

where $t_f$ is calculated as $(r_A + r_O)/(\sqrt{2}(r_B + r_O))$ and $r_A$, $r_B$, and $r_O$ are the weighted-average Shannon's ionic radii[53] of the A-site, B-site, and O atom, respectively, in the perovskite lattice. A similar procedure was used to determine the values for other features for each composition.

The objective of classification learning is to build ML models that map the features to the labels (+1 or −1). We use the support vector classifier with a radial basis function kernel (SVC$_{rbf}$) for building the classification learning models (see Methods section for details). We first evaluated the suitability of the features and the SVC$_{rbf}$ ML method on our initial dataset by splitting the dataset into training and independent test sets,

because very little work has been reported in the literature on predicting the perovskite phase stability of these solid solutions. We generated 100 training and test set pairs via random sampling, such that each training and test set pair contained 127 and 40 data points, respectively. We built 100 SVC$_{rbf}$ classifiers from the training sets and used them to predict the labels for the compositions in the corresponding test sets. We calculated the accuracy of each SVC$_{rbf}$ classifier by evaluating its performance on each of the test sets. The overall accuracy was then estimated by calculating the mean value from the outcome of the 100 SVC$_{rbf}$ classifiers. We find that our SVC$_{rbf}$ classifiers performed with an average accuracy of 77.5 ± 6.4%, which is reasonable and not uncommon in the literature[12,15,54], if dealing with heterogeneous experimental datasets such as the one constructed in this work. This exercise provides an estimate of the predictive power of the features and the SVC$_{rbf}$ method in classifying +1 and −1 labels of the compositions assembled in our dataset.

For the prediction of new perovskites in the unexplored 61,506 composition space, however, we utilized all 167 data points for training the SVC$_{rbf}$ classifiers. We constructed a total of 100 bootstrapped samples[55,56] from the original dataset and built 100 SVC$_{rbf}$ classifiers. Thus, any given composition will have predictions of either +1 or −1 label from 100 SVC$_{rbf}$ classifiers. Since the objective is to down-select promising perovskites from the 61,506 unexplored composition space, we "exploit" our SVC$_{rbf}$ classifiers, i.e., we only choose those compositions that are classified in the +1 (or desired) label at least 95 or more times. The hyperparameters for the SVC$_{rbf}$ classifiers were optimized by the tenfold cross-validation (CV) method (see Methods section for details).

**Regression and active learning**. We built another dataset of 117 compositions for which the $T_C$ data are known from published experiments. This dataset contains compositions that are both at and away from the Morphotropic Phase Boundary (MPB) composition, but we do not distinguish between them. In the ferroelectrics literature, the term MPB refers to structural phase transitions arising due to changes in chemical composition at a given temperature and especially in PbTiO$_3$-based materials, MPB encompasses a region in the phase diagram where two ferroelectric phases (typically in tetragonal and rhombohedral symmetries) coexist. The $T_C$ data were determined using the dielectric measurements in an impedance analyzer. For this dataset, we represent each composition using two features, namely $t_f$ and ionic displacement. Unlike the classification learning problem, the rationale for the choice of these features in predicting the $T_C$ for Bi(Me′Me″)O$_3$ − PT solid solutions is well-established in the literature[20,42–44,57–59]. For instance, Abrahams et al.[59] showed that in displacive ferroelectrics, $T_C \propto \delta^2$, where $\delta$ is the relative atomic displacement of the homopolar metal atom from the center of the octahedron cage. Later, Grinberg et al.[42,57,58] extended the argument to also include displacements from the A-site atoms. Similarly, Eitel et al.[20] showed the existence of a correlation between $t_f$ and $T_C$ at the MPB compositions in BiMeO$_3$-PT solid solutions, where Me is a trivalent octahedral cation. However, the uncertainties associated with $t_f$ and $\delta$ in describing the $T_C$ at compositions away from the MPB are not known. The goal of regression is to build ML models that can predict $T_C$ as a function of $t_f$, $\delta$, and compositions (at and away from the MPB). While the $t_f$ for each composition is calculated as shown in Eq. (1), the $\delta$ for the solid solution is calculated as follows:

$$\delta_{\text{solid solution}} = \{x \times \delta_{\text{Bi}} \times \delta_{\text{Me′Me″}}\} + \{(1 − x) \times \delta_{\text{Pb}} \times \delta_{\text{Ti}}\}, \quad (2)$$

where the values for $\delta$ for each octahedral site cation (Ti, Me′, and

**Table 1 List of ten new compositions predicted and validated by ML and experimental measurements**

| Iteration # | Compositions | Perovskites (+1 or −1) | | Curie temperature $T_C$ (K) | | EGO or Exploitation | c/a |
| | | Prediction from $SVC_{rbf}$ | XRD measurements | Prediction from $SVR_{rbf}$ ($\widehat{T_C} \pm \sigma$) | Dielectric measurements | | |
|---|---|---|---|---|---|---|---|
| 1 | $0.6Bi(Sc_{0.17}Ga_{0.83})O_3$–0.4PT | +1 | −1 | 747 ± 223 | – | EGO | – |
| 2a | $0.35Bi(In_{0.31}Sc_{0.69})O_3$–0.65PT | +1 | +1 | 923 ± 343 | 733 | EGO | 1.029 |
| 2b | $0.5Bi(Sc_{0.56}Ga_{0.44})O_3$–0.5PT | +1 | −1 | 846 ± 229 | – | EGO | – |
| 3a | $0.25Bi(Sc_{0.47}Ga_{0.53})O_3$–0.75PT | +1 | +1 | 778 ± 135 | 798 | EGO | 1.058 |
| 3b | $0.7Bi(Fe_{0.73}Co_{0.27})O_3$–0.3PT | +1 | −1 | 929 ± 94 | – | Exploitation | – |
| 4a | $0.4Bi(Fe_{0.19}Sc_{0.81})O_3$–0.6PT | +1 | +1 | 749 ± 161 | 728 | EGO | 1.026 |
| 4b | $0.2Bi(Fe_{0.12}Co_{0.88})O_3$–0.8PT | +1 | +1 | 815 ± 24 | 898 | Exploitation | 1.066 |
| 5a | $0.3Bi(Yb_{0.44}Al_{0.56})O_3$–0.7PT | +1 | −1 | 801 ± 125 | – | EGO | – |
| 5b | $0.3Bi(Ni_{0.50}Sn_{0.50})O_3$–0.7PT | +1 | +1 | 780 ± 123 | 658 | EGO | 1.027 |
| 5c | $0.2Bi(Co_{0.90}Al_{0.10})O_3$–0.8PT | +1 | +1 | 843 ± 33 | 883 | Exploitation | 1.063 |

EGO or Exploitation within the active learning strategy recommended the corresponding composition for synthesis. $SVC_{rbf}$ and $SVR_{rbf}$ are the classification learning and regression ML methods, respectively. Labels +1 and −1 in the XRD measurements column refer to compositions in the desired perovskite phase and those that contained undesired secondary phases, respectively. The XRD and dielectric measurements provide the ground truth for validating the $SVC_{rbf}$ and $SVR_{rbf}$ predictions. The tetragonality (c/a ratio) from XRD measurements are also given. In the column named "Iteration", we use a, b, or c as indicators to only uniquely identify compositions and they carry no scientific information

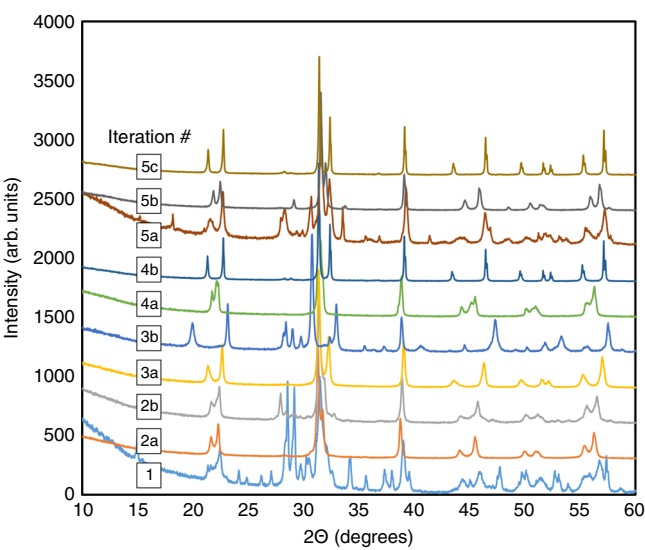

**Fig. 4** X-ray diffraction data for the ten compositions. Compositions synthesized at iterations 2a, 3a, 4a, 4b, 5b, and 5c are identified to be mostly perovskites (≥95% phase pure), in agreement with the predictions from $SVC_{rbf}$ ML method. See Table 1 for details

Me″) were taken from the work of Balachandran et al.[60] and that for Bi and Pb comes from Grinberg and Rappe[58]. All $\delta$ values used in this work are given in Supplementary Table 1. In the case of $\delta_{Me'Me''}$, we used the weighted-average values for the individual $\delta_{Me'}$ and $\delta_{Me''}$ cation data.

We used the support vector regression with the radial basis function kernel ($SVR_{rbf}$) for building the regression models. Similar to classification learning, we construct a total of 100 bootstrapped samples from the original dataset and build 100 $SVR_{rbf}$ models. The mean value and standard deviation from these 100 $SVR_{rbf}$ models was taken as the mean $T_C$ prediction $\left(\widehat{T_C}\right)$ and the associated uncertainty ($\sigma$), respectively. Once the $SVR_{rbf}$ models are built, we apply them to predict $\widehat{T_C} \pm \sigma$ for the candidate perovskites that were down-selected from the classification learning. The hyperparameters for the $SVR_{rbf}$ models were optimized by the tenfold CV method (see Methods section for details). The performance of the $SVR_{rbf}$ on the training set is shown in Fig. 3. The mean absolute error on the initial dataset was estimated to be 30.2 K.

We explored two methods to rank the candidate perovskites for experimental recommendations. One is based on the EGO[61], where we calculate the "expected improvement, $E(I)$" for each unmeasured composition using the expression, $\sigma[\phi(z) + z\Phi(z)]$, where $z = \left(\widehat{T_C} - \mu^*\right)/\sigma$ and $\mu^*$ is the maximum $T_C$ observed so far in the current training set, $\phi(z)$ and $\Phi(z)$ are the standard normal density and cumulative distribution functions, respectively[61]. Here, $E(I)$ balances the tradeoff between "exploitation" and "exploration" of the $SVR_{rbf}$ model. At the end of each iteration, EGO returns a score for $E(I)$ for each unmeasured composition, whose relative magnitude depends on the $SVR_{rbf}$ predicted $\left(\widehat{T_C}, \sigma\right)$ pair for those compositions and the value of $\mu^*$ in the training set. It is common to pick the composition with the maximum $E(I)$ for validation and feedback. It is anticipated that the validation of compositions recommended by $E(I)$ would lead to model improvement in the subsequent iterations. The second method, in contrast to the EGO method, exploits the $SVR_{rbf}$ models, i.e., we recommend candidate perovskite compositions from the unexplored space that were predicted to have the largest $\widehat{T_C}$. Unlike $E(I)$, these recommendations are not expected to improve the $SVR_{rbf}$ models in the subsequent iterations. We refer to this as "Exploitation" in this paper.

**Experiments**. All specimens were prepared by the conventional powder processing technique. The general procedure for the processing used in this work is outlined in a previous publication by Kowalski et al.[62–64]. However, the sintering temperature was varied between 1173 and 1373 K due to the dependence of melting temperature on compositions. During sintering the weight loss was less than 3% for all compositions. A sintered pellet from each composition was then crushed for XRD. All compositions studied were, to some degree, perovskite. In general, there is a solubility limit depending on the cations in $Bi(Me'Me'')O_3$ and this limit tends to be less than 50% except for $Bi(Zn_{0.5}Ti_{0.5})O_3$ and $BiFeO_3$ in PT. If XRD revealed the composition to be mostly perovskite (≥95%), then a sintered pellet was prepared for dielectric measurements (see Methods for additional details).

**Iterative loop**. The iterative loop consists of the following steps: (a) Screen 61,506 compositions using an ensemble of 100 $SVC_{rbf}$ classifiers and down-select compositions as candidate perovskites, (b) Predict the $\widehat{T_C}$ along with the uncertainties ($\sigma$) for the down-selected candidates using $SVR_{rbf}$ regression models, (c) Calculate the $E(I)$ for each candidate perovskite using the EGO algorithm

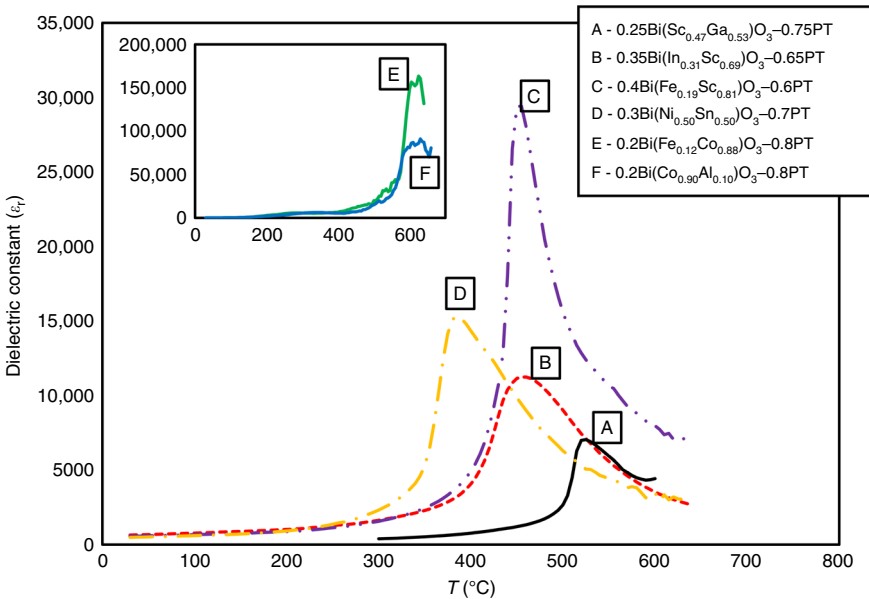

**Fig. 5** Dielectric data for the six confirmed perovskites. Dielectric constant ($\varepsilon_r$) as a function of temperature (T, in °C) data measured at 10 kHz frequency for the six experimentally confirmed perovskites (labeled A–F, whose compositions are also provided alongside the data). We put the data for E and F compositions in the inset because the samples become conductive. The dielectric constant at that phase transformation is not real. It is mostly due to charge arising from conduction

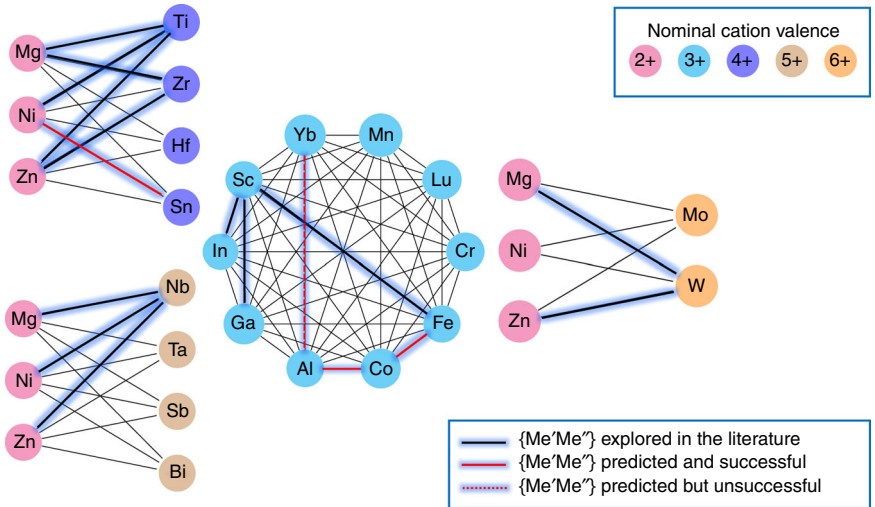

**Fig. 6** {Me′Me″}-pair combinations in the chemical space. In the Bi(Me′Me″)O₃-PT chemical space, where there are 23 cations (of different nominal valence) that can potentially occupy the Me′ or Me″ sites, we have a total of 75 possible {Me′Me″} pairs (thin black lines). However, only 13 such pairs (highlighted in dark blue line) are experimentally explored in the literature and are included in our training set. In this work, we predict and experimentally synthesize four new pairs (highlighted in red line) that are identified as potential high-temperature ferroelectric perovskites by ML. Among them, three pairs (highlighted in full red line)—{FeCo}, {NiSn}, and {CoAl}—are experimentally found to form in the perovskite structure and are identified as promising systems for further investigation. One of the pairs (dotted red line), {YbAl}, was predicted to be a promising high $T_C$ perovskite by ML, but XRD revealed presence of secondary phases (therefore, unsuccessful)

and rank them in the descending order, (d) Recommend the promising compositions [i.e., the one with the largest E(I) for experimental synthesis and characterization, (e) Synthesize and evaluate the phase stability using XRD and only those that are revealed to be perovskite are prepared for measuring the $T_C$, and (f) Augment the training set with these compositions and repeat the cycle until our budget of ten experiments is expended. The results are summarized in Table 1, where we also provide the data for tetragonality (c/a ratio) of the experimentally confirmed perovskites. A detailed account describing how the search process evolved from one iteration to another is given in Supplementary

Note 1 and we only discuss our main results here. In Figs. 4 and 5, we show the XRD and dielectric measurements data for the synthesized compositions, respectively. From XRD measurements, we identify six new compositions that were synthesized in iteration 2a, 3a, 4a, 4b, 5b, and 5c to be mostly perovskite (≥95% phase pure). The temperature-dependent dielectric measurements were then performed only on the six compositions that are confirmed as perovskites by the XRD data.

Among the six experimentally confirmed perovskite compositions, 0.2Bi(Fe₀.₁₂Co₀.₈₈)O₃–0.8PT (predicted in the fourth iteration using the exploitation method) had the highest

measured $T_C$ of 898 K. Both $Fe^{3+}$ and $Co^{3+}$ cations have large ionic displacements in the perovskite lattice[60], which is hypothesized as one of the main reasons for its large $T_C$. Our work has led to the prediction of four new promising {Me'Me''} pairs in the chemical space, namely {FeCo}, {NiSn}, {CoAl}, and {YbAl}, not explored in the literature. This is shown in Fig. 6 and the exact compositions are given in Table 1. The XRD measurements confirmed three of them, {FeCo}, {NiSn}, and {CoAl}, in the perovskite structure. In the case of the {YbAl} pair, XRD measurements revealed secondary phases and therefore, our preliminary studies identify them as unsuccessful.

## Discussion

One of the major hurdles in traditional materials design has been the vast size of the unexplored space and our experimental bias towards knowledge of the well-known materials systems such as $BaTiO_3$, $Pb(Zr, Ti)O_3$ and $Pb(Mg, Nb)O_3$-$PbTiO_3$. We have demonstrated here how synthesis can be guided by a two-step ML strategy that learns from all available data, including both successful and failed experiments, to accelerate the search for new high-$T_C$ perovskite oxides. Our work has led to two promising compositions, $0.2Bi(Fe_{0.12}Co_{0.88})O_3$–$0.8PT$ and $0.2Bi(Co_{0.90}Al_{0.10})O_3$–$0.8PT$. The $Bi(FeCo)O_3$-PT and $Bi(CoAl)O_3$-PT chemical spaces rank third and fourth highest, respectively, in terms of their measured $T_C$ values compared to the state-of-the-art compositions [$BiFeO_3$-PT and $Bi(ZnTi)O_3$-PT] in our training set. The merits of these compositions are their high $T_C$ and the ease of processing in the perovskite phase, which are critical for actuator performance. This is important because the operation temperature is limited by AC conductivity, leakage, or de-poling and $T_C$ sets the intrinsic limit to piezoelectricity. In future, one needs to incorporate $c/a$ ratio and domain mobility, which we have ignored in this work.

From Table 1, we find two intriguing observations pertaining to the $BiGaO_3$-$BiScO_3$-PT and $BiFeO_3$-$BiCoO_3$-PT predicted compositions that are worth discussing. In $BiGaO_3$-$BiScO_3$-PT, the first two predictions (in iterations 1 and 2, as given in Table 1) did not result in a perovskite phase as revealed by the XRD data. However, the third prediction (in iteration 3) resulted in perovskite structure. Notice that the composition predicted in iteration 3 had less $BiGaO_3$ content (13.25%) relative to the first two predictions (that contained 49.8 and 22% $BiGaO_3$). The PT content, on the other hand, also increased in the third prediction compared to the first two predictions (Table 1). Similarly, in $BiFeO_3$-$BiCoO_3$-PT the first prediction (in iteration 3, $0.7Bi(Fe_{0.73}Co_{0.27})O_3$–$0.3PT$) was rich in $Bi(CoFe)O_3$ relative to that of the PT content. Our XRD measurements revealed secondary phases and therefore, unsuccessful. In the next iteration (# 4), ML predicted $0.2Bi(Fe_{0.12}Co_{0.88})O_3$–$0.8PT$, which is now rich in PT compared to the $Bi(CoFe)O_3$ content. The XRD measurements revealed perovskite phase. We interpret these outcomes as the $SVC_{rbf}$ classifier iteratively "learning" the solubility limit of $Bi(Me'Me'')O_3$ in PT perovskite from previous failed experiments.

We also shed some light on the evolution of the ML models as a result of ten experiments. For instance, the $SVR_{rbf}$ models (after iteration 4) showed only marginal improvement relative to that of the first set of models (before iteration 1). The mean absolute error value decreased from 30 to 28 K. On the other hand, the $SVC_{rbf}$ classification learning models showed no improvement in its performance on the test set. We calculated an average accuracy of 77 ± 6.5% for the final model (i.e., after iteration 4), which is almost identical to that of the initial $SVC_{rbf}$ models (before iteration 1), whose average accuracy was calculated as 77.5 ± 6.4%. We attribute this behavior to the exploitative

recommendation strategy of the $SVC_{rbf}$ models. We anticipate that the average accuracies of these classifiers can, in principle, be further improved when these models are coupled to active learning methods, such as uncertainty sampling[15], where a loss function can be defined with an intention of improving the average accuracies of the classification learning models. We identify them as promising directions for future investigation.

## Methods

**Machine learning**. We use the C-support vector classification (C-SVC) and $\varepsilon$-support vector regression ($\varepsilon$-SVR) methods with non-linear Gaussian radial basis function kernel for classification learning and regression, respectively, as implemented in the e1071 package[65] within the RSTUDIO environment[66]. The hyperparameters (cost and gamma) for both C-SVC and $\varepsilon$-SVR methods were optimized using the tenfold CV method and the grid search method. In the tenfold CV method, the original dataset was divided randomly (without replacement) into ten subsamples of equal sizes. Then, a single subsample (containing 10% of the data) was chosen as the test set and the remaining 90% was used for training the models. The process was repeated ten times (one for each subsample) such that each composition appear exactly once in the test set data. The set of hyperparameters that resulted in the smallest tenfold CV error (i.e., classification error in the case of C-SVC and mean squared error in the case of $\varepsilon$-SVR) was chosen for training our final models. Bootstrap resampling is one of the well-known methods, where a dataset is sampled repeatedly with replacement[55,56].

**Experimental**. The starting oxide powders used and compositions studied are listed here, $PbO$, $TiO_2$, $Bi_2O_3$, $Fe_2O_3$, $CoO$, $Ga_2O_3$, $In_2O_3$, $Al_2O_3$, $Sc_2O_3$, $SnO_2$, $NiO$, and $Yb_2O_3$. The purity of the powders used exceeds 99%.

The disc pellets were first parallel polished to maintain a 10:1 diameter to thickness ratio. Then the polished pellets were electroded with silver paint and cured at 350 °C. To then investigate the prediction of the $T_C$, dielectric constant measurements were carried out as a function of temperature at 0.5 V with an Agilent 4294A impedance analyzer (Agilent Technologies, Santa Clara, CA) in tandem with a customized CM furnace. Data was captured through a computer interface by LABVIEW (National Instruments Corp., Austin, TX) software.

**Data availability**. The datasets generated and analyzed for classification learning and regression are deposited at figshare[67] with identifier https://doi.org/10.6084/m9.figshare.5687551.v1.

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

## Acknowledgements

P.V.B. and T.L. acknowledge funding support from the Center for Nonlinear Science (CNLS) and the LDRD project #20140013DR at Los Alamos National Laboratory (LANL). B.K. and A.S. were supported by the Air Force Office of Scientific Research (AFOSR) Grant #FA9550-0601-1-0260.

## Author contributions

The study was planned and the manuscript prepared by P.V.B., B.K., T.L., and A.S. Machine learning studies was performed by P.V.B and T.L. Experiments were performed by B.K and A.S. All authors discussed the results, wrote, and commented on the manuscript.

## Additional information

**Competing interests:** The authors declare no competing interests.

