## [Peer Review File · Nature Communications]

Reviewer #1 (Remarks to the Author):

The authors use machine learning (support vector classification and regression) to search for candidate ferroelectric perovskite materials with a high ferroelectric Curie temperature. The paper is well written and the research is well designed. A positive aspect of the article is the use of experimental data, rather than computational data, to train the machine learning models. I recommend publication in Nature Communications after the following minor comments are addressed.

1) The authors make heavy use of abbreviations and acronyms throughout the article. I would suggest that they double-check to make sure all are well defined. For example, the acronym "MPB", though widely used in the field, is not defined in the article.

2) It would be helpful for the authors to describe how their results compare to the state-of-the-art and similar compounds. Are Curie temperatures for the materials they have discovered significantly better or worse than competing materials? Or are they comparable?

3) The authors state that they have ignored the tetragonality ratio and domain mobility. Why? It would seem that it would be straightforward to calculate the tetragonality ratio unless I am missing something. Even if the values aren't ideal, it would be valuable to the community to know what they are.

Reviewer #2 (Remarks to the Author):

The authors report an approach for theory-guided materials synthesis as applied to ferroelectric perovskites. They utilize machine learning on a known body of materials to establish relationship between materials structural and compositional descriptors and Curie temperature, and further utilize this knowledge (and associated uncertainties) to guide experimental synthesis. This data-rich experiment-limited approach is ideally suited for classical ceramics synthesis routines, which generally do not allow fast sampling of large parameter spaces, and will be of interest for a broad community of solid state chemists. Furthermore, I believe the same approach can be further extended towards oxide film growth, where substrate temperature, laser fluency, and gas pressure offer additional parameters.

Overall, I am very impressed by this work, and recommend it for publications

February 6, 2018

Letter to the Referees

Note: Our response to referees' comments are given in purple text color.

Report of the First Referee – NCOMMS-17-32548

Response to Referee #1

Comment: The authors use machine learning (support vector classification and regression) to search for candidate ferroelectric perovskite materials with a high ferroelectric Curie temperature. The paper is well written and the research is well designed. A positive aspect of the article is the use of experimental data, rather than computational data, to train the machine learning models. I recommend publication in Nature Communications after the following minor comments are addressed.

Response: We thank the referee for his/her comments on our paper and recommending it for publication. We use this letter to provide point-by-point response to his/her comments.

Comment: The authors make heavy use of abbreviations and acronyms throughout the article. I would suggest that they double-check to make sure all are well defined. For example, the acronym “MPB”, though widely used in the field, is not defined in the article.

Response: We appreciate the comment. We have carefully reviewed our manuscript and made revisions to address this point. We have tried our best to minimize the usage of abbreviations and where appropriate, we have attempted to define them clearly. We have added a definition for the morphotropic phase boundary (MPB) on Page 4, which now reads as follows:

“We built another dataset of 117 compositions for which the T_C data are known from published experiments. This dataset contains compositions that are both at and away from the Morphotropic Phase Boundary (MPB) composition, but we do not distinguish between them. In the ferroelectrics literature, the term MPB refers to structural phase transitions arising due to changes in chemical composition at a given temperature and especially in PbTiO_3 based materials, MPB encompasses a region in the phase diagram where two ferroelectric phases (typically in tetragonal and rhombohedral symmetries) coexist.”

We first refer to Efficient Global Optimization (EGO) on Page 2 and now added the following text on Page 2 for clarification,

“We also use our recently developed active learning (or adaptive design) approach based on efficient global optimization (EGO)^{13,17} to recommend promising compositions for experimental synthesis and characterization (details of EGO algorithm are discussed in the Regression and Active Learning section).”

Comment: It would be helpful for the authors to describe how their results compare to the state-of-the-art and similar compounds. Are Curie temperatures for the materials they have discovered significantly better

or worse than competing materials? Or are they comparable?

Response: The highest Curie temperature (T_C) in our training dataset belongs to the $\text{BiFeO}_3\text{-PbTiO}_3$ solid solutions ($T_C = 1101$ K) and the second highest belongs to the $\text{Bi}(\text{Zn}_{1/2}\text{Ti}_{1/2})\text{O}_3\text{-PbTiO}_3$ ($T_C = 990$ K) solid solutions. These chemical spaces are well studied in the literature. On the other hand, our two new solid solutions, namely $\text{Bi}(\text{Fe}_{0.12}\text{Co}_{0.88})\text{O}_3\text{-PbTiO}_3$ and $\text{Bi}(\text{Al}_{0.10}\text{Co}_{0.90})\text{O}_3\text{-PbTiO}_3$, rank third ($T_C = 898$ K) and fourth ($T_C = 883$ K), respectively, relative to the known materials. More importantly, the merits of these compositions are their ease of processing in the perovskite phase (in addition to having a reasonably high T_C), which are critical for actuator performance. Thus, we qualify these materials as competitive (at-best) relative to the state-of-the-art. Furthermore, our work is one of the first in identifying $\text{Bi}(\text{FeCo})\text{O}_3$ and $\text{Bi}(\text{AlCo})\text{O}_3$ end members as promising high- T_C materials that can also be stabilized in the *desired* perovskite phase. We anticipate that these results will spur new activities on these interesting materials class. We have added a sentence on Page 6 (first paragraph in Section Discussion) in our manuscript, which now reads as follows:

“The $\text{Bi}(\text{FeCo})\text{O}_3\text{-PT}$ and $\text{Bi}(\text{CoAl})\text{O}_3\text{-PT}$ chemical spaces rank third and fourth highest, respectively, in terms of their measured T_C values compared to the state-of-the-art compositions [$\text{BiFeO}_3\text{-PT}$ and $\text{Bi}(\text{ZnTi})\text{O}_3\text{-PT}$] in our training set.”

Comment: The authors state that they have ignored the tetragonality ratio and domain mobility. Why? It would seem that it would be straightforward to calculate the tetragonality ratio unless I am missing something. Even if the values aren’t ideal, it would be valuable to the community to know what they are.

Response: This is a good question. There are two ways to formulate the problem,

- **Use tetragonality or domain mobility as an independent variable (similar to how we considered tolerance factor in this paper).** One of our key requirements for building machine learning models is that we should be able to quantify our input data such that they not only “fingerprint” known compositions but also represent the yet-to-be-explored composition space. Since descriptors such as tetragonality and domain mobility are not known to us *a priori* before performing experiments (for the $\sim 61,500$ compositions), we cannot incorporate them as inputs to our machine learning. Therefore, we ignored them in our machine learning study. On the other hand, we can represent each of the $\sim 61,500$ composition using tolerance factor (as it is not an experimental outcome) and hence, tolerance factor is a good independent variable for our problem.
- **Use tetragonality or domain mobility as a dependent variable or unknown (similar to T_C).** We can, in principle, formulate a multiobjective optimization problem such that we are in search of a new composition with a high T_C AND within a desired tetragonality ratio (e.g., 1–1.07) AND domain mobility. However, we do not take this route in this work. This is mainly because we do not have the capability (yet) to handle multiobjective optimization design problems (optimize two or more continuous variables such as T_C , tetragonality and domain mobility). What we have shown in this paper is the potential of using machine learning methods for *predicting both phase stability and optimization of a physical property (T_C)* based on learning from experimental data. It should also be noted that this paper is one of the first in the literature to demonstrate these ideas using experiments. Clearly, constrained optimization is the path forward in this research (which we also state in the manuscript, See **Discussion** section). We are in the process of developing machine learning methods for multiobjective optimization problems, such as those desired by this referee.

Report of the Second Referee – NCOMMS-17-32548

Response to Referee #2

Comment: The authors report an approach for theory-guided materials synthesis as applied to ferroelectric perovskites. They utilize machine learning on a known body of materials to establish relationship between materials structural and compositional descriptors and Curie temperature, and further utilize this knowledge (and associated uncertainties) to guide experimental synthesis. This data-rich experiment-limited approach is ideally suited for classical ceramics synthesis routines, which generally do not allow fast sampling of large parameter spaces, and will be of interest for a broad community of solid state chemists. Furthermore, I believe the same approach can be further extended towards oxide film growth, where substrate temperature, laser fluency, and gas pressure offer additional parameters.

Overall, I am very impressed by this work, and recommend it for publications

Response: We thank the referee for his/her interest in our work and recommending it for publication.

Reviewer #1 (Remarks to the Author):

The authors have adequately addressed nearly all of my concerns. Regarding the omission of the tetragonality ratio and domain mobility, it is understandable why they would leave these out of their machine learning model. However it would be helpful to include these values for the new compositions listed in Table 1, so that the reader can better assess the merits of the new compositions. Of course it would be appropriate to explain that these values have not been optimized by the machine learning model and are only provided to give the readers more information about the new compositions.

February 28, 2018

LETTER TO THE REFEREE

Note: Our response to referees' comments are given in purple text color.

REPORT OF THE FIRST REFEREE – NCOMMS-17-32548

Response to Referee #1

Comment: The authors have adequately addressed nearly all of my concerns. Regarding the omission of the tetragonality ratio and domain mobility, it is understandable why they would leave these out of their machine learning model. However it would be helpful to include these values for the new compositions listed in Table 1, so that the reader can better assess the merits of the new compositions. Of course it would be appropriate to explain that these values have not been optimized by the machine learning model and are only provided to give the readers more information about the new compositions.

Response: We thank the referee for his/her comments on our paper and we have incorporated the tetragonality (c/a ratio) for the new compositions in Table 1. Our updated Table 1 on Page 6 now reads as follows,

TABLE 1 | List of ten new compositions predicted and validated by ML and experimental measurements. EGO or Exploitation within the active learning strategy recommended the corresponding composition for synthesis. SVC_{rbf} and SVR_{rbf} are the classification learning and regression ML methods, respectively. Labels +1 and -1 in the XRD measurements column refer to compositions in the desired perovskite phase and those that contained undesired secondary phases, respectively. The XRD and dielectric measurements provide the ground truth for validating the SVC_{rbf} and SVR_{rbf} predictions. The tetragonality (\$c/a\$ ratio) from XRD measurements are also given. In the column named "Iteration", we use a, b, or c as indicators to only uniquely identify compositions and they carry no scientific information.

Iteration #	Compositions	Perovskites (+1 or -1)		Curie temperature T_C (K)		EGO or Exploitation	c/a
		Prediction from SVC_{rbf}	XRD measurements	Prediction from SVR_{rbf} ($\hat{T}_C \pm \sigma$)	Dielectric measurements		
1	0.6Bi($Sc_{0.17}Ga_{0.83}$) O_3 -0.4PT	+1	-1	747 ± 223	-	EGO	-
2a	0.35Bi($In_{0.31}Sc_{0.69}$) O_3 -0.65PT	+1	+1	923 ± 343	733	EGO	1.029
2b	0.5Bi($Sc_{0.56}Ga_{0.44}$) O_3 -0.5PT	+1	-1	846 ± 229	-	EGO	-
3a	0.25Bi($Sc_{0.47}Ga_{0.53}$) O_3 -0.75PT	+1	+1	778 ± 135	798	EGO	1.058
3b	0.7Bi($Fe_{0.73}Co_{0.27}$) O_3 -0.3PT	+1	-1	929 ± 94	-	Exploitation	-
4a	0.4Bi($Fe_{0.19}Sc_{0.81}$) O_3 -0.6PT	+1	+1	749 ± 161	728	EGO	1.026
4b	0.2Bi($Fe_{0.12}Co_{0.88}$) O_3 -0.8PT	+1	+1	815 ± 24	898	Exploitation	1.066
5a	0.3Bi($Yb_{0.44}Al_{0.56}$) O_3 -0.7PT	+1	-1	801 ± 125	-	EGO	-
5b	0.3Bi($Ni_{0.50}Sn_{0.50}$) O_3 -0.7PT	+1	+1	780 ± 123	658	EGO	1.027
5c	0.2Bi($Co_{0.90}Al_{0.10}$) O_3 -0.8PT	+1	+1	843 ± 33	883	Exploitation	1.063